# Social Determinants of Health and 30-Day Readmission for Heart Failure Patients in U.S. Hospitals: Evidence from ICD-10 Z-Code Data

**DOI:** 10.3390/healthcare13172102

**Published:** 2025-08-23

**Authors:** Yong Cai, Liu Yanping, Qiang Liu

**Affiliations:** 1IQVIA (United States), 2400 Ellis Road, Durham, NC 27703, USA; yocai@csumb.edu (Y.C.); ypliu@us.imshealth.com (L.Y.); 2Daniels School of Business, Purdue University, West Lafayette, IN 47907, USA

**Keywords:** social determinants of health (SDoHs), heart failure (HF), readmission, Z-codes, health equity

## Abstract

Background/Objectives: There has been growing interest in understanding the impact of social determinants of health (SDoHs) on health outcomes. Since 2015, healthcare providers have begun to document patients’ SDoH systematically using ICD-10 Z-codes. Methods: We extracted claims data from a nationally representative hospital chargemaster database for 586,929 eligible HF patients between January 2019 and December 2021. We investigated the association between SDoH Z-codes and 30-day hospital readmission for heart failure (HF) patients in U.S. hospitals using a Chi square test and adjusted odds ratios from logistic regression models. Results: We found that four major SDoH Z-code categories and five specific sub-Z-code factors within the major categories are significantly associated with 30-day readmission for HF patients. We also found that patients with two or more SDoH Z-codes have a higher risk of readmission than those with one. Conclusions: Our study indicates that ICD-10 Z-codes are useful in identifying SDoH risk factors for hospital readmission among HF patients. Policymakers and healthcare providers should consider Z-codes when assessing HF readmission risk and developing interventions to lower HF readmission rates.

## 1. Introduction

Heart failure (HF) remains one of the leading causes of hospital readmission in the United States [1]. Reducing avoidable hospital readmissions has been a focus of efforts to improve the quality of patient care and lower healthcare costs. Despite some improvements, the 30-day hospital readmission rate for heart failure (HF) patients has been reported to be over 20% [2], and more than half of health systems pay penalties for readmission, including among HF patients [3]. These readmissions not only reflect challenges in post-discharge care and chronic disease management, but also contribute to avoidable healthcare expenditures [4,5].

Research has shown that up to 86% of HF rehospitalizations may be preventable through medical and social interventions [6]. A growing body of literature has emphasized that clinical risk factors alone cannot fully explain readmission risk in HF patients [7,8]. Social determinants of health (SDoHs)—including socioeconomic status, housing stability, education, employment, and social support—are increasingly recognized as critical influences on patient outcomes, including readmission [9,10]. For instance, housing insecurity may reduce a patient’s ability to attend follow-up visits, afford medications, or adhere to lifestyle changes required for HF management [11,12]. Similarly, social isolation or social frailty may hinder adherence to treatment plans and contribute to disease progression [13,14]. In 2020, the American Hospital Association published a scientific statement by White-Williams et al. to encourage more research on the impact of SDoH on HF patients [15].

The existing literature on social determinants of health (SDoHs) and heart failure (HF) outcomes largely focuses on sociodemographic factors such as gender, race, and age, while giving much less attention to socioeconomic variables like income and education, and often overlooking social environmental factors such as housing stability and social support [16]. To support the identification of non-medical risk factors, the Centers for Medicare & Medicaid Services (CMS) and the American Hospital Association (AHA) have encouraged the use of ICD-10-CM Z-codes [17] to systematically document SDoH in clinical records. Z-codes provide a standardized way to record patient-level SDoH data across a wide range of domains. However, adoption of Z-codes remains inconsistent across providers and hospitals, and their value in predicting patient outcomes—such as 30-day hospital readmission—remains underexplored in large-scale studies.

The objective of this study is to examine whether—and which—SDoHs, as captured by ICD-10 Z-codes, are significantly associated with 30-day all-cause hospital readmission among heart failure (HF) patients in U.S. hospitals. Leveraging nationally representative chargemaster claims data, we assess the predictive value of Z-codes at both the category and sub-code levels, while adjusting for patient demographics, clinical history, and hospital-level characteristics. To the best of our knowledge, this is the first study to investigate the association between ICD-10 Z-coded SDoHs and HF readmissions using a nationally representative patient claims dataset. By identifying the most salient SDoH indicators linked to readmission risk, our findings may inform healthcare providers and policymakers about which social risk factors should be prioritized when designing interventions aimed at reducing preventable readmissions. Additionally, the use of national data improves the generalizability of our findings and addresses limitations of earlier studies based on regional data or single health systems.

## 2. Methods

### 2.1. Data Source

We use IQVIA’s Charge Data Master (CDM) as the primary data source for this research. The CDM is an accounting system specific to each hospital that tracks the details of its in and outpatient visits from all pay types. IQVIA’s CDM data are collected from more than 900 short-term, general non-federal acute care hospitals. The key metrics include patient diagnoses, procedures and tests, and medicine and devices. Our research covers an overall three-year study period from January 2019 to December 2021.

We supplement the CDM with IQVIA medical claims (Dx) and longitudinal prescription (LRx) data to track patients’ complete medical histories outside hospitals. Dx captures patient-level diagnoses, procedures, and in-office treatments for physician office visits, covering 75% of AMA providers and collecting 1.5 billion claims annually. LRx tracks patient-level prescriptions, covering 92% of data from the retail channel, 72% from the mailing channel, and 76% from the long-term care (LTC) channel. All data are bridgeable by an anonymous patient ID and Health Insurance Portability and Accountability Act (HIPAA) compliant. The data are subjected to a series of quality checks to ensure a standardized format and to minimize error rates.

### 2.2. Hospital and Patient Eligibility and Selection

To avoid bias from data reporting issues, we define eligible hospital panels as those continuously supplying monthly data during our study period. We further focused on hospitals with an SDoH Z-code reporting rate larger than or equal to the median value. Since AHA published the new guidelines for reporting Z-codes in 2018, allowing all clinicians, not just physicians, to report on these SDoH Z-codes [18], the hospital Z-code reporting rate has constantly been increasing. However, not all hospitals report Z-codes consistently. From 2019 to 2021, the median Z-code reporting rates in the hospital data were 0.33%, 0.41%, and 0.48%, respectively. If we include data from hospitals that do not report SDoH Z-codes or under-report Z-codes, our statistical analysis based on sample pooling across hospitals may severely underestimate the impact of SDoH Z-codes on 30-day readmissions. Therefore, we focus on hospitals that consistently reported SDoHs over time and had reporting rates above the median. It is important to acknowledge that focusing only on hospitals with above-median Z-code reporting may limit the generalizability of our findings to hospitals with more established SDoH screening and documentation practices. Hospitals with lower Z-code reporting rates may differ systematically in terms of patient population, resource availability, or institutional priorities related to social risk screening. As such, the associations we report may not fully capture the SDoH-readmission dynamics in hospitals with underdeveloped or inconsistent Z-code adoption.

Patients with at least one HF-related inpatient visit from the eligible hospitals were selected for three yearly cohorts. The complete HF diagnosis codes used for patient selection are listed in Appendix A. For each patient, adjacent hospitalizations were combined as a single stay, and the index hospitalization was their first HF-related inpatient stay during the selection period. Patient samples were further refined based on the following criteria: patients had to be discharged alive, have valid data for region, age, gender, and payer type, and have continuous enrolment in medical and pharmacy benefits during the 12 months before and one month after the index hospitalization. We also included only patients treated at hospitals with valid characteristic information, including bed size. The final sample contains 586,929 unique patients with an index heart failure hospitalization during the study period. Of these patients, 32,181 (5.5%) have at least one SDoH Z-code recorded in the 1-year pre-index period. Appendix A provides more details on patient selection and exclusion.

Data preprocessing included cleaning to resolve inconsistencies and ensure completeness. Due to strict patient selection criteria, there were no missing values for hospital and patient demographics or Charles Comorbidity Index (CCI) scores (range: 0–33). Aside from CCI and patient age, the only remaining numeric variable—prior heart failure hospitalizations in the past year—was recoded into three categories: 0, 1, and 2 or more.

### 2.3. Statistical Analysis

In this study, we define readmission as hospitalization for any cause within 30 days following the discharge of the index hospitalization. We created the patient-level medical history covariates, including CCI, based on claims activities before the HF hospitalization. The patient demographic covariates include age, gender, region, and payer type. Three hospital characteristics (bed size, teaching hospital yes/no, and hospital location (rural vs. urban)) were also included as covariates. There are ten SDoH Z-code categories and 76 sub-Z-codes in ICD-10. These Z-codes capture patients’ education, housing, economic circumstances, social environment, physical environment, transportation difficulties, and other non-medical factors [19]. Although our models include clinical and demographic controls such as the CCI, age, and payer type, we acknowledge that other social confounders not captured by Z-codes—such as insurance instability or mental health conditions—may also influence readmission risk and warrant further investigation.

We first conducted Chi-square independence tests [20] to assess the association between each major SDoH Z-code category and 30-day hospital HF patient readmission preliminarily. We then developed a series of multivariate logistic regression models—referred to as individual models—to evaluate the impact of each Z-code category on readmission separately. To assess the joint effect of all SDoH Z-code categories, we also constructed a multivariate logistic regression model that included all categories simultaneously, referred to as the full model. Both individual and full models controlled for patient medical history, demographic characteristics, and hospital-level factors. Odds ratios and *p*-values were reported to quantify the associations between SDoH Z-codes and readmission. We compared the odds ratios of Z-code categories across the individual and full models.

Additionally, we fitted separate full models for sub-codes within each major Z-code category. We also applied stepwise logistic regression to identify significant predictors among all 76 sub-Z-codes for HF patient readmission. Lastly, we estimated a logistic model using the number of unique SDoH Z-code categories per patient (categorized as 0, 1, or ≥2) as predictors to evaluate the cumulative burden of social risk. All *p*-values were calculated using two-sided Wald statistics.

## 3. Results

### 3.1. Baseline Characteristics

Characteristics of patients with or without SDoH Z-codes are shown in Table 1. As expected, the HF patient sample distribution skewed toward older patients, with 69.5% equal to or above 65. Among all the eligible HF patients, Medicare (58.2%) and commercially insured (38.0%) are the dominant insurance pay types. Between patients with and without SDoH Z-codes, the distribution of previous HF-related hospitalizations, previous HF diagnoses, and CCI scores is significantly different (*p* < 0.05). As previously mentioned, the patient characteristics shown in Table 1 were included as covariates in the subsequent multivariate logistic models.

### 3.2. Chi-Square Association Test

We conducted Chi-square tests to examine whether each SDoH Z-code category was significantly associated with 30-day all-cause hospital readmission, without control for other covariates. These tests provide preliminary model-free evidence for the association between SDoH Z-codes and 30-day all-cause hospital readmission. Table 2 reports patient SDoH incidents, readmission counts, and *p*-values from the Chi-square tests. Eight of the ten SDoH Z-code categories are significantly associated with readmission (*p* <= 0.001). The two insignificant categories are occupational exposure risk (Z57.xx) and environmental pollution (Z77.1x). Among all SDoH Z-codes, housing and economics are the most common problems, with a prevalence rate of 2.35%, followed by the social environment, which has a rate of 1.36%.

### 3.3. SDoH Adjusted Odds Ratios

Table 3 and Figure 1 and Figure 2 present the adjusted odds ratios and statistical significance for each SDoH Z-code category from the individual and full models. The estimation results show a similar estimated odds ratio between the two models. In the following discussions, we will focus on the full model results.

All covariates for patient medical baseline controls are significantly associated with 30-day hospital readmission (see Appendix A). Patient age, gender, region, pay type, hospital bed size, and the index hospitalization year are all statistically significant.

Among all the SDoH Z-code variables included in this study, Z59.xx housing and economic (OR = 1.45; *p* < 0.005), Z56.xx employment (OR = 1.23; *p* < 0.005), Z60.xx social environment (OR = 1.13; *p* < 0.005), and Z63.xx primary support group (OR = 1.11; *p* = 0.01) are the four most significant factors associated with 30-day all-cause hospital readmission. Z62.xx upbringing (OR = 1.16; *p* = 0.06) and Z55.xx education and literacy (OR = 1.24; *p* = 0.09) also show significance at a *p*-value level of 0.1.

To investigate the details of each Z-code category, we examined the distribution of the sub-codes within each category and built separate multivariate logistic regression models, including all the sub-codes within each category and controlling for the same covariates. Table 4 and Figure 3 describe the sub-code distributions, adjusted odds ratios, and *p*-values for the category of housing and economic circumstances (Z59.xx), which has the highest risk of 30-day hospital readmission. Homelessness (Z59.0) is the most common problem, accounting for 75.3% of all housing and economic difficulties. Homelessness is also a significant risk factor (*p* < 0.005) associated with 30-day all-cause hospital readmission, with the highest odds ratio (OR = 1.62) in the category. Other significant sub-codes include lack of adequate food (Z59.4; OR = 1.43; *p* = 0.02), other problems related to housing and economic circumstances (Z59.8x; OR = 1.24; *p* = 0.01), and problems related to housing and economic circumstances unspecified (Z59.9; OR = 1.17; *p* = 0.02).

Table 5 and Figure 4 describe the statistics of the problems related to the employment (Z56.xx) Z-code category. A total of 96.4% of the problems reported are related to unemployment unspecified (Z56.0), which is also a significant factor (OR = 1.35; *p* < 0.005). Another significant problem is unspecified problems related to employment (Z56.9; OR = 2.31; *p* = 0.01), but it is less common and only accounts for 1.5% of all reported problems within the category.

The statistics of the sub-Z-codes for the Z60.x social environment category are shown in Table 6 and Figure 5.Within the category, problems related to living alone (Z60.2) are the most reported problem, which account for 88.5% of all cases. Z60.2 is also a significant factor associated with 30-day readmission (OR = 1.15; *p* < 0.005). The other less common but significant problems include problems related to the social environment unspecified (Z60.9; 4.4%; OR = 1.49; *p* < 0.005) and other problems related to the social environment (Z60.8; 2.1%; OR = 1.44; *p* < 0.005).

For the primary support group (Z63.xx) category, the distribution of the subcodes is more spread out, as shown in Table 7 and Figure 6. The most reported problem (Z63.4 disappearance and death of family member) accounts for less than half of all cases (42.1%), and it is only significant with a *p*-value level of 0.1 (OR = 1.11; *p* = 0.09). There are five significant problems within the category: Z63.5 disruption of the family by separation and divorce (11.3%; OR = 1.47; *p* < 0.005), Z63.8 other specified problems related to a primary support group (19.3%; OR = 1.25; *p* = 0.01), Z63.9 problem related to primary support group unspecified (8.0%; OR = 1.37; *p* = 0.02), Z63.7x other stressful life events affecting family and household (9.9%; OR = 1.29; *p* = 0.03), and Z63.1 problems in the relationship with in-laws (0.4%; OR = 3.67; *p* = 0.02).

Within the other two categories (Z62.xxx upbringing; Z55.x education literacy), which are significantly associated with 30-day all-cause readmission with a *p*-value level of 0.1, personal history of abuse in childhood (Z62.81x; 79.6%; OR = 1.43; *p* < 0.005) and education literacy unspecified (Z55.9; 33.2%; OR = 1.75; *p* = 0.01) are the two significant problems. Additional statistics of the sub-Z-codes within these two categories can be found in Appendix A, respectively, in the Appendix A.

### 3.4. Association Between All SDoH Sub-Z-Codes and Readmission

To examine the association between all the sub-Z-codes and HF readmission, we fitted a stepwise multivariate logistic regression model to select the most relevant factors from the large number of sub-Z-codes (totaling 76). We also control for patient and hospital characteristics as before. The stepwise model outputs are listed in Appendix A. Table 8 and Figure 7 show the adjusted odds ratio and *p*-value of the sub-Z-codes retained in the final stepwise model. There are sixteen sub-Z codes kept in the final model. Among those sub-Z-codes, twelve across six Z-code categories are significant at 0.05. Excluding those with low prevalence (reported cases < 100) and unspecified subcategories, the model identified five specific sub-Z codes significantly associated with 30-day all-cause readmission. They are Z59.0 homelessness (1.77%; OR = 1.60; *p* < 0.005); Z63.5 disruption of the family by separation and divorce (0.07%; OR = 1.34; *p* = 0.01); Z56.0 unemployment (0.43%; OR = 1.22; *p* < 0.005); Z62.81x personal history of abuse in childhood (0.10%; OR = 1.22; *p* = 0.03); Z60.2 problems related to living alone (1.20%; OR = 1.14; *p* < 0.005). An interesting finding is that low income (Z59.6) has a negative association with HF readmission (OR = 0.87; *p* = 0.08), while other problems all have a positive association.

### 3.5. Association Between the Number of SDoH Codes and Readmission

One may wonder whether the number of SDoH Z-codes a HF patient has is associated with an increased risk of hospital readmission for the patient. To answer this question, we fitted a patient-level multivariate logistic model. We included two dummy variables to capture patients’ SDoH status at the major category level: having no SDoH Z-code, having one SDoH Z-code, and having more than one SDoH Z-code. In the study sample, 29,308 patients recorded only one SDoH problem, and 2873 patients had two or more SDoH codes. As shown in Table 9 and Figure 8, patients from the group with one SDoH code have a 1.23 OR of readmission (95% CI: 1.20–1.26, *p* < 0.005), while patients with two or more SDoH codes show a much higher OR of 1.62 (95% CI: 1.50–1.75, *p* < 0.005).

## 4. Discussion

In 2018, the American Hospital Association (AHA) published the guidance allowing all clinicians, not limited to physicians, to report SDoH ICD-10 Z-codes for patients [17]. Since this clarification, the utilization of SDoH Z-codes in medical claims has increased [21]. However, so far, little research has been performed to evaluate the predictive value of using Z-codes to assess the impact of SDoH on health outcomes. We investigated the association between Z-codes and 30-day readmission for HF patients using the IQVIA hospital chargemaster claims database, which is nationally representative and covers all age patients.

We found that, among the ten major SDoH Z-code categories, four were significantly associated with 30-day hospital readmission for HF patients: Z59.xx housing and economic, Z56.xx unemployment, Z60.xx social environment, and Z63.xx primary support group. In addition, Z62.xx upbringing and Z55.xx education and literacy are significant at a *p*-value of 0.1 in the fully adjusted multivariate regression model. Surprisingly, psychosocial (Z64.xx-Z65.xx), occupational risk (Z57.xx), environmental pollution (Z77.1), and transportation problems (Z91.89) do not appear to be significantly associated with HF readmission in our study.

### 4.1. Pathway and Intervention

Although the mechanisms and pathways by which SDoH impacts health outcomes could be complex and challenging to study [22], much has been learned over the years on the value of SDoH for understanding patient risk and identifying intervention opportunities to improve outcomes. For example, Pendyal et al. found that lack of intensive self-management and post-discharge care can impede HF recovery and that homelessness and housing instability pose a barrier to HF self-management [11]. Our results show that housing instability (OR = 1.45; *p* < 0.005) is the most significant SDoH risk factor for HF readmission. Within the category, homelessness is the most reported and significant issue (75.3%; OR = 1.60; *p* < 0.005). This is consistent with the findings by Doran et al. that homeless patients had strikingly high 30-day hospital readmission rates. In their chart review study, 74.8% of homeless patients were readmitted within two weeks [12].

Employment is one of the common factors studied in SDoH research [23]. Our results show that employment-related problems (Z56.x) pose a 23% greater risk of HF readmission (OR = 1.23; 95% CI: 1.13–1.34; *p* < 0.005) and unemployment (Z56.0) is the biggest issue (96.4%) within the category, posing a 22% greater risk of HF readmission (OR = 1.22; 95% CI: 1.12–1.33; *p* < 0.005). This supports previously proposed solutions by White-Williams et al., which included attempts to improve and sustain employment and programs to mitigate CV risk factors to reduce health inequalities for HF patients who are unemployed [15].

Gorij et al. conducted a meta-analysis of the impact of social isolation on HF patient readmission rates [13]. Social isolation was defined as living alone, having a poor social network, and having little social contact. Their study reported that social isolation was associated with a 55% increase in hospital readmission risk among HF patients (OR = 1.55; 95% CI 1.39–1.73; *p* < 0.005). Other related research also showed social environment as a significant risk factor for HF readmission in various patient populations [14,24]. In our analyses using Z-codes, we have found that problems related to the social environment (Z60.xx) pose a 13% greater risk of HF readmission (OR = 1.13; 95% CI: 1.08–1.19; *p* < 0.005). We also found that the problems related to living alone (Z60.2) sub-Z-code (OR = 1.14; 95% CI: 1.08–1.20; *p* < 0.005) pose a 14% greater risk, and the problems related to social environment unspecified (Z60.9) sub-Z-code (OR = 1.29; 95% CI: 1.02–1.62; *p* = 0.03) pose a 29% greater risk of HF readmission.

This research indicates that family circumstances and primary support are significant factors that impact HF patient readmission. Disturbing family events, such as separation and divorce (OR = 1.34; 95% CI: 1.08–1.66; *p* < 0.01), and other stressful life events that affect families and households, are top risk factors. The healthcare providers shall pay special attention to HF patients who experience such events.

Our results also show that patients with ≥2 SDoH Z-codes have a much higher 30-day HF readmission risk (OR = 1.62; 95% CI: 1.50–1.75; *p* < 0.005) than those with only one SDoH Z-code (OR = 1.23; 95% CI: 1.20–1.26; *p* < 0.005). It is reasonable to speculate that patients with multiple SDoH issues represent an underprivileged population with greater socioeconomic risk and more difficulties managing post-discharge HF self-care. However, Sterling and colleagues found that HF patients with a higher number of SDoH (≥2) did not exhibit a higher risk of 90-day mortality than those with only one SDoH [25]. There are some explanations for this discrepancy between the two studies. First, the outcome measures are different. Sterling et al. measured mortality, while our study measured hospital readmission. Second, different types of SDoH metrics were used in calculating the number of SDoH. There is only a small overlap of SDoH categories between the two studies. For example, the Sterling et al. study included SDoH factors such as black race, health infrastructure, etc., while our study included employment problems, transportation difficulty, etc. Given these contradictory findings, additional research is needed to better understand the risks associated with multiple SDoH factors.

These effect sizes are not only statistically significant but also clinically meaningful. For instance, the observed odds ratio of 1.60 for homelessness (Z59.0) suggests a 60% increase in the odds of 30-day readmission, a magnitude comparable to or greater than many clinical risk factors used in current HF risk models. Similarly, the 1.62 odds ratio for patients with two or more SDoH Z-codes indicates a compounded social burden that substantially raises readmission risk. These results underscore the importance of integrating SDoH data into clinical risk assessments and tailoring interventions—such as housing support, care coordination, or social work referrals—for patients with elevated social risk profiles.

While our quantitative findings provide evidence of the association between specific SDoH and HF readmissions, the real-world implementation of Z-code documentation remains limited. In clinical practice, Z-codes are often underutilized due to workflow constraints, lack of provider training, limited EHR integration, and inconsistent institutional priorities. To improve adoption, health systems could embed standardized SDoH screening tools into intake processes, offer provider incentives, and integrate Z-code reporting into quality improvement initiatives. Moreover, the observed discrepancies between our findings and those of Sterling et al.—where we found strong associations between multiple SDoH and readmission but not mortality in their study—highlight the need to align SDoH measurement with outcome-specific pathways. Readmission risk may be more sensitive to environmental and behavioral stressors (e.g., housing, family disruption), while mortality may be more influenced by biological severity. Qualitative studies examining how SDoH factors interact with patient care navigation, self-management, and provider response could further illuminate these differences and inform intervention design.

Although our study focuses on demonstrating the value and validity of using ICD-10 Z-codes to understand SDoH impacts on HF patient hospital readmission, researchers can likely extend similar applications of Z-codes to other disease areas. Researchers, healthcare providers, and policymakers could benefit from additional insights by including Z-codes data in their SDoH studies.

#### Limitations

ICD-10 Z-codes have limitations related to the documentation of SDoH problems. First, clinicians may have different interpretations of Z-codes, and multiple Z-codes are available to cover similar SDoH problems [26]. Second, there is no direct Z-code available for transportation difficulties. Some hospitals and providers advocate using ‘Z91.89 Other specified risk factors, not elsewhere classified’ for patient transportation problems [17], but there is no guarantee that all providers follow the same coding convention. Finally, some HF patients’ SDoH issues may not have been recorded in the study sample, potentially biasing our estimates. While our approach improves data validity by minimizing measurement error from non-reporting hospitals or under-reporting hospitals, future research is warranted to assess how SDoH impacts readmission in under-reporting hospitals and to explore methods for adjusting or imputing missing SDoH data.

## 5. Conclusions

This study contributes to the growing body of evidence demonstrating the impact of social determinants of health (SDoHs), as captured by ICD-10 Z-codes, on HF outcomes. Using a large national dataset in the United States and a robust modeling strategy, we show that patients with documented SDoH Z-codes have significantly higher odds of 30-day readmission. While some adjusted odds ratios appear modest in size, their clinical relevance is underscored by the prevalence of these risk factors and their potential for intervention at the system level.

Importantly, our findings reveal that certain SDoH domains—particularly housing and economic instability, social environment, and employment—are more strongly associated with readmission risk. Compared to studies focusing on other outcomes (e.g., mortality in Sterling et al.), our analysis highlights the value of outcome-specific modeling when evaluating the role of SDoHs in clinical care.

Despite growing awareness of the importance of social determinants of health, Z-code documentation remains inconsistently used in clinical practice. Our findings reinforce the need for standardized coding guidelines and clearer definitions of qualifying social conditions to reduce ambiguity and support provider decision-making. Institutional efforts to embed SDoH screening into clinical workflows—alongside clearer policy incentives and alignment with quality metrics—may be necessary to improve uptake. Enhancing the consistency and accuracy of Z-code reporting will ultimately strengthen population health analytics and support interventions aimed at reducing preventable HF readmissions.

## Figures and Tables

**Figure 1 healthcare-13-02102-f001:**
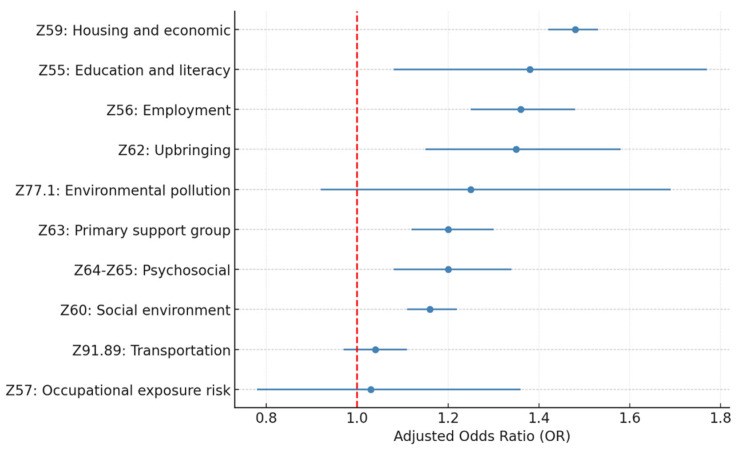
Adjusted OR for 30-day HF readmission by SDoH Z-code category (individual model).

**Figure 2 healthcare-13-02102-f002:**
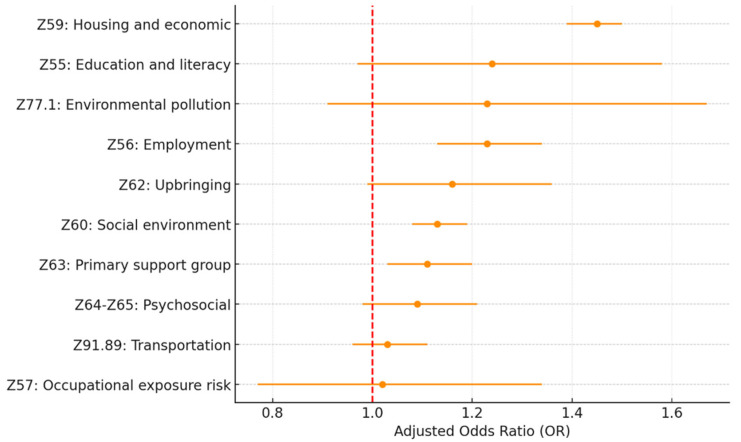
Adjusted OR for 30-day HF readmission by SDoH Z-code Category (Full model). Source: Authors’ analysis of IQVIA claim data between January 2019 and December 2021.

**Figure 3 healthcare-13-02102-f003:**
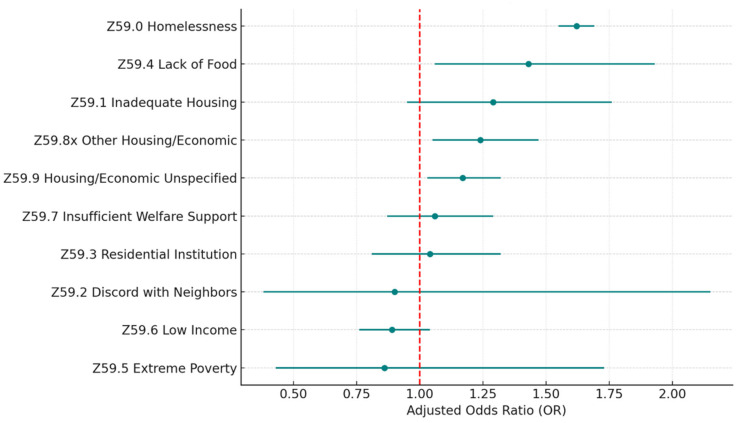
Z59 housing and economic SDoH (sub-Z-codes). Source: Authors’ analysis of IQVIA claim data between January 2019 and December 2021.

**Figure 4 healthcare-13-02102-f004:**
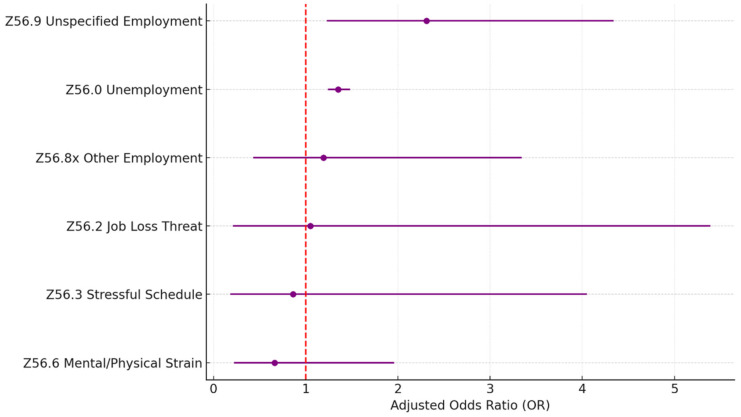
Z56 housing and economic SDoH (sub-Z-codes). Source: Authors’ analysis of IQVIA claim data between January 2019 and December 2021.

**Figure 5 healthcare-13-02102-f005:**
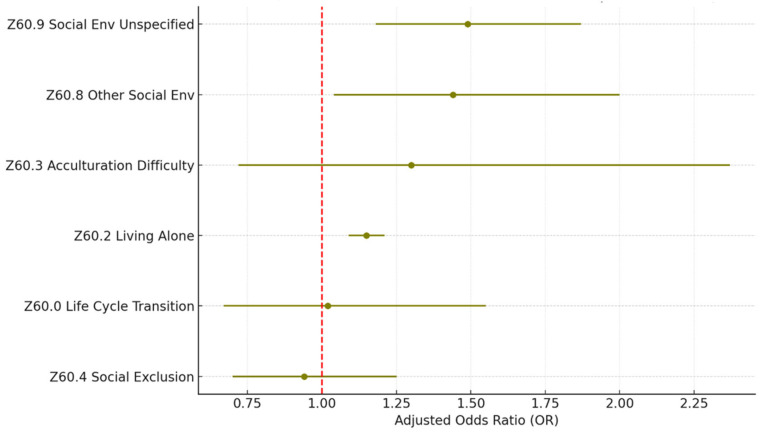
Z60 social environment SDoH (sub-Z-codes). Source: Authors’ analysis of IQVIA claim data between January 2019 and December 2021.

**Figure 6 healthcare-13-02102-f006:**
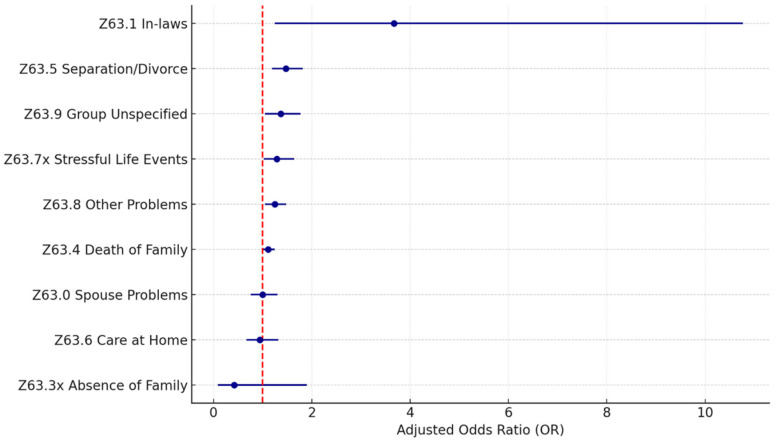
Z63 primary support group SDoH (sub-Z-codes). Source: Authors’ analysis of IQVIA claim data between January 2019 and December 2021.

**Figure 7 healthcare-13-02102-f007:**
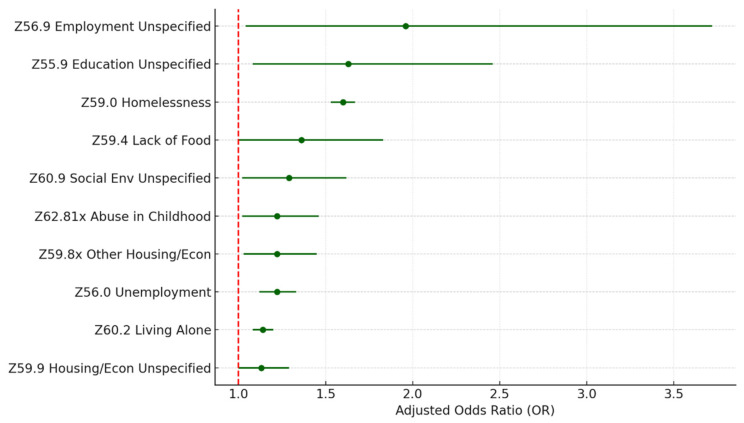
Top sub-Z-codes in stepwise model for readmission. Source: Authors’ analysis of IQVIA claim data between January 2019 and December 2021.

**Figure 8 healthcare-13-02102-f008:**
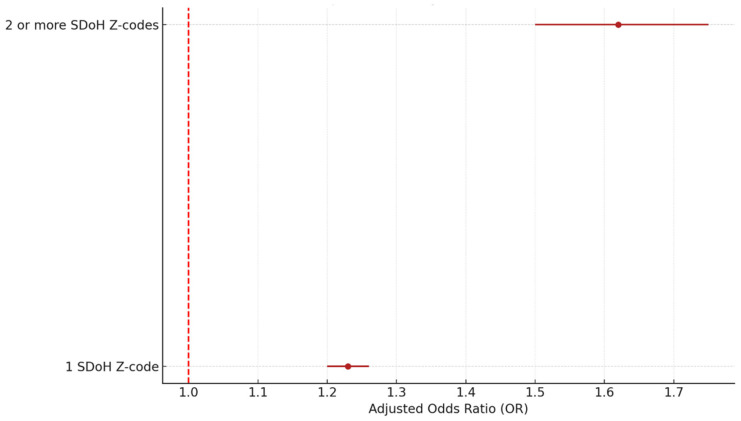
Adjusted OR by number of SDoH categories. Source: Authors’ analysis of IQVIA claim data between January 2019 and December 2021.

**Table 1 healthcare-13-02102-t001:** Patient baseline characteristics for patients with and without SDoH.

Characteristics	Total *N* = 586,929	No SDOH Z-Code *N* = 554,748	Any SDOH Z-Codes *N* = 32,181	*p* Value *
Age Group				<0.001
<18	3135 (0.5%)	2895 (0.5%)	240 (0.7%)	
18–34	9818 (1.7%)	8836 (1.6%)	982 (3.1%)	
35–44	18,745 (3.2%)	16,869 (3.0%)	1876 (5.8%)	
45–54	44,529 (7.6%)	40,053 (7.2%)	4476 (13.9%)	
55–64	102,727 (17.5%)	94,365 (17.0%)	8362 (26.0%)	
65+	407,975 (69.5%)	391,730 (70.6%)	16,245 (50.5%)	
Number of1-year previous HF hospitalizations				<0.001
0	441,124 (75.2%)	420,818 (75.9%)	20,306 (63.1%)	
1	53,859 (9.2%)	50,460 (9.1%)	3399 (10.6%)	
2+	91,946 (15.7%)	83,470 (15.0%)	8476 (26.3%)	
Charles Comorbidity Index (CCI)				0.012
0–2	93,043 (15.9%)	87,765 (15.8%)	5278 (16.4%)	
3–5	211,457 (36.0%)	200,024 (36.1%)	11,433 (35.5%)	
6+	282,429 (48.1%)	266,959 (48.1%)	15,470 (48.1%)	
Gender				<0.001
Female	280,001 (47.7%)	266,134 (48.0%)	13,867 (43.1%)	
Male	306,928 (52.3%)	288,614 (52.0%)	18,314 (56.9%)	
Previous HF diagnosis				<0.001
No	266,128 (45.3%)	254,258 (45.8%)	11,870 (36.9%)	
Yes	320,801 (54.7%)	300,490 (54.2%)	20,311 (63.1%)	
Region				<0.001
MIDWEST	65,644 (11.2%)	60,753 (11.0%)	4891 (15.2%)	
NORTHEAST	97,976 (16.7%)	92,766 (16.7%)	5210 (16.2%)	
SOUTH	230,751 (39.3%)	221,197 (39.9%)	9554 (29.7%)	
WEST	192,558 (32.8%)	180,032 (32.5%)	12,526 (38.9%)	
Payer Type				<0.001
CASH	1765 (0.3%)	1632 (0.3%)	133 (0.4%)	
COMMERCIAL	222,770 (38.0%)	208,270 (37.5%)	14,500 (45.1%)	
MEDICAID	20,723 (3.5%)	18,357 (3.3%)	2366 (7.4%)	
MEDICARE	341,671 (58.2%)	326,489 (58.9%)	15,182 (47.2%)	
Hospital Location:				<0.001
Rural	50,546 (8.6%)	47,333 (8.5%)	3213 (10.0%)	
Urban	536,383 (91.4%)	507,415 (91.5%)	28,968 (90.0%)	
Hospital Teaching Flag				<0.001
No	298,831 (50.9%)	283,164 (51.0%)	15,667 (48.7%)	
Unknown	78,149 (13.3%)	74,634 (13.5%)	3515 (10.9%)	
Yes	209,949 (35.8%)	196,950 (35.5%)	12,999 (40.4%)	
Hospital Bed Size				<0.001
1–99 beds	24,495 (4.2%)	23,036 (4.2%)	1459 (4.5%)	
100–199 beds	92,844 (15.8%)	87,399 (15.8%)	5445 (16.9%)	
200–299 beds	105,955 (18.1%)	99,893 (18.0%)	6062 (18.8%)	
300–499 beds	174,764 (29.8%)	165,042 (29.8%)	9722 (30.2%)	
500 or more beds	188,871 (32.2%)	179,378 (32.3%)	9493 (29.5%)	
Index HF Hospitalization Year				<0.001
2019	201,638 (34.4%)	192,469 (34.7%)	9169 (28.5%)	
2020	192,485 (32.8%)	181,616 (32.7%)	10,869 (33.8%)	
2021	192,806 (32.8%)	180,663 (32.6%)	12,143 (37.7%)	

* *p*-value based on chi-square statistic. Source: Authors’ analysis of IQVIA claim data between January 2019 and December 2021.

**Table 2 healthcare-13-02102-t002:** Chi-square test of association between SDOH and heart failure 30-day readmission.

	Total (*N* = 586,929)	No 30-Day Readmission (*N* = 453,833)	All-Cause 30-Day Readmission (*N* = 133,096)	*p* Value
Z55.xx: Education and literacy	316 (0.05%)	220 (0.05%)	96 (0.07%)	0.001
Z56.xx: Employment	2646 (0.45%)	1822 (0.40%)	824 (0.62%)	<0.001
Z57.xx: Occupational exposure risk	279 (0.05%)	210 (0.05%)	69 (0.05%)	0.454
Z59.xx: Housing and economic	13,811 (2.35%)	9444 (2.08%)	4367 (3.28%)	<0.001
Z60.xx: Social environment	7954 (1.36%)	5787 (1.28%)	2167 (1.63%)	<0.001
Z62.xx: Upbringing	758 (0.13%)	527 (0.12%)	231 (0.17%)	<0.001
Z63.xx: Primary support group	3510 (0.60%)	2550 (0.56%)	960 (0.72%)	<0.001
Z64.xx-Z65.xx: Psychosocial	1737 (0.30%)	1249 (0.28%)	488 (0.37%)	<0.001
Z77.1x: Environmental pollution	215 (0.04%)	155 (0.03%)	60 (0.05%)	0.080
Z91.89: Transportation	4398 (0.75%)	3273 (0.72%)	1125 (0.85%)	<0.001

Source: Authors’ analysis of IQVIA claim data between January 2019 and December 2021.

**Table 3 healthcare-13-02102-t003:** SDOH Z-codes adjusted the odds ratio of 30-day hospital readmission for heart failure.

	Individual SDoH Adjusted Odds Ratio	*p* Value	Full Model SDoH Adjusted Odds Ratio	*p* Value
Z55.xx: Education and literacy	1.38 (1.08, 1.77)	0.01	1.24 (0.97, 1.58)	0.09
Z56.xx: Employment	1.36 (1.25, 1.48)	0.00	1.23 (1.13, 1.34)	0.00
Z57.xx: Occupational exposure risk	1.03 (0.78, 1.36)	0.82	1.02 (0.77, 1.34)	0.89
Z59.xx: Housing and economic	1.48 (1.42, 1.53)	0.00	1.45 (1.39, 1.50)	0.00
Z60.xx: Social environment	1.16 (1.11, 1.22)	0.00	1.13 (1.08, 1.19)	0.00
Z62.xx: Upbringing	1.35 (1.15, 1.58)	0.00	1.16 (0.99, 1.36)	0.06
Z63.xx: Primary support group	1.20 (1.12, 1.30)	0.00	1.11 (1.03, 1.20)	0.01
Z64.xx-Z65.xx: Psychosocial	1.20 (1.08, 1.34)	0.00	1.09 (0.98, 1.21)	0.12
Z77.1: Environmental pollution	1.25 (0.92, 1.69)	0.15	1.23 (0.91, 1.67)	0.18
Z91.89: Transportation	1.04 (0.97, 1.11)	0.29	1.03 (0.96, 1.11)	0.37
Log-Likelihood			−3.08 × 10^5^	
LLR *p*-value			0.000	
AUC			0.6013	

Source: Authors’ analysis of IQVIA claim data between January 2019 and December 2021.

**Table 4 healthcare-13-02102-t004:** Z59.xx Housing and economic SDoH odds ratio of 30-day hospital readmission for heart failure.

	N (%)	Subcodes Adjusted Odds Ratio	*p* Value
Z59.0 Homelessness	10,394 (75.3%)	1.62 (1.55, 1.69)	0.00
Z59.9 Problem Related to Housing and Economic Circumstances Unspecified	1227 (8.9%)	1.17 (1.03, 1.32)	0.02
Z59.6 Low Income	887 (6.4%)	0.89 (0.76, 1.04)	0.15
Z59.8x Other Problems Related to Housing and Economic Circumstances	680 (4.9%)	1.24 (1.05, 1.47)	0.01
Z59.7 Insufficient Social Insurance and Welfare Support	539 (3.9%)	1.06 (0.87, 1.29)	0.59
Z59.3 Problems Related to Living in Residential Institution	364 (2.6%)	1.04 (0.81, 1.32)	0.77
Z59.4 Lack of Adequate Food	200 (1.4%)	1.43 (1.06, 1.93)	0.02
Z59.1 Inadequate Housing	195 (1.4%)	1.29 (0.95, 1.76)	0.10
Z59.5 Extreme Poverty	45 (0.3%)	0.86 (0.43, 1.73)	0.67
Z59.2 Discord with Neighbors Lodgers and Landlord	29 (0.2%)	0.90 (0.38, 2.15)	0.82

Source: Authors’ analysis of IQVIA claim data between January 2019 and December 2021.

**Table 5 healthcare-13-02102-t005:** Z56.xx Employment SDoH odds ratio of 30-day hospital readmission for heart failure.

	N (%)	Subcodes Adjusted Odds Ratio	*p* Value
Z56.0 Unemployment Unspecified	2552 (96.4%)	1.35 (1.24, 1.48)	0.00
Z56.9 Unspecified Problems Related to Employment	41 (1.5%)	2.31 (1.23, 4.34)	0.01
Z56.6 Other Physical and Mental Strain Related to Work	25 (0.9%)	0.66 (0.22, 1.96)	0.46
Z56.8x Other Problems Related to Employment	20 (0.8%)	1.19 (0.43, 3.34)	0.74
Z56.3 Stressful Work Schedule	11 (0.4%)	0.86 (0.18, 4.05)	0.85
Z56.2 Threat of Job Loss	8 (0.3%)	1.05 (0.21, 5.39)	0.95
Z56.5 Uncongenial Work Environment	1 (0.0%)	NA	NA
Z56.1 Change of Job	0 (0.0%)	NA	NA
Z56.4 Discord with Boss and Workmates	0 (0.0%)	NA	NA

Source: Authors’ analysis of IQVIA claim data between January 2019 and December 2021.

**Table 6 healthcare-13-02102-t006:** Z60.x social environment SDoH odds ratio of 30-day hospital readmission for heart failure.

	N (%)	Subcodes Adjusted Odds Ratio	*p* Value
Z60.2 Problems Related to Living Alone	7042 (88.5%)	1.15 (1.09, 1.21)	0.00
Z60.9 Problem Related to Social Environment Unspecified	347 (4.4%)	1.49 (1.18, 1.87)	0.00
Z60.4 Social Exclusion and Rejection	271 (3.4%)	0.94 (0.70, 1.25)	0.66
Z60.8 Other Problems Related to Social Environment	171 (2.1%)	1.44 (1.04, 2.00)	0.03
Z60.0 Problems of Adjustment to Life Cycle Transitions	120 (1.5%)	1.02 (0.67, 1.55)	0.92
Z60.3 Acculturation Difficulty	53 (0.7%)	1.30 (0.72, 2.37)	0.38
Z60.5 Target of Perceived Adverse Discrimination and Persecution	2 (0.0%)	2.94 (0.18, 47.89)	0.45

Source: Authors’ analysis of IQVIA claim data between January 2019 and December 2021.

**Table 7 healthcare-13-02102-t007:** Z63.xx primary support group SDoH odds ratio of 30-day hospital readmission for heart failure.

	N (%)	Subcodes Adjusted Odds Ratio	*p* Value
Z63.4 Disappearance and Death of Family Member	1479 (42.1%)	1.11 (0.98, 1.25)	0.09
Z63.8 Other Specified Problems Related to Primary Support Group	676 (19.3%)	1.25 (1.05, 1.48)	0.01
Z63.5 Disruption of Family by Separation and Divorce	395 (11.3%)	1.47 (1.19, 1.82)	0.00
Z63.7x Other Stressful Life Events Affecting Family and Household	348 (9.9%)	1.29 (1.02, 1.64)	0.03
Z63.0 Problems in Relationship with Spouse or Partner	300 (8.5%)	1.00 (0.76, 1.30)	0.99
Z63.9 Problem Related to Primary Support Group Unspecified	280 (8.0%)	1.37 (1.05, 1.77)	0.02
Z63.6 Dependent Relative Needing Care at Home	200 (5.7%)	0.94 (0.67, 1.32)	0.72
Z63.3x Absence of Family Member	15 (0.4%)	0.42 (0.09, 1.90)	0.26
Z63.1 Problems in Relationship with In-Laws	14 (0.4%)	3.67 (1.25, 10.77)	0.02

Source: Authors’ analysis of IQVIA claim data between January 2019 and December 2021.

**Table 8 healthcare-13-02102-t008:** SDoH sub-Z-codes stepwise model adjusted odds ratio of 30-day hospital readmission for heart failure.

	N (%)	Stepwise SDoH Subcodes Adjusted Odds Ratio	*p* Value
Z55.9 Education Literacy Unspecified	105 (0.02%)	1.63 (1.08, 2.46)	0.02
Z56.0 Unemployment Unspecified	2552 (0.43%)	1.22 (1.12, 1.33)	0.00
Z56.9 Unspecified Problems Related to Employment	41 (0.01%)	1.96 (1.04, 3.72)	0.04
Z59.0 Homelessness	10,394 (1.77%)	1.60 (1.53, 1.67)	0.00
Z59.1 Inadequate Housing	195 (0.03%)	1.25 (0.92, 1.70)	0.15
Z59.4 Lack of Adequate Food	200 (0.03%)	1.36 (1.00, 1.83)	0.05
Z59.6 Low Income	887 (0.15%)	0.87 (0.74, 1.02)	0.08
Z59.8x Other Problems Related to Housing and Economic Circumstances	680 (0.12%)	1.22 (1.03, 1.45)	0.02
Z59.9 Problem Related to Housing and Economic Circumstances Unspecified	1227 (0.21%)	1.13 (1.00, 1.29)	0.05
Z60.2 Problems Related to Living Alone	7042 (1.20%)	1.14 (1.08, 1.20)	0.00
Z60.9 Problem Related to Social Environment Unspecified	347 (0.06%)	1.29 (1.02, 1.62)	0.03
Z62.81x Personal History of Abuse in Childhood	603 (0.10%)	1.22 (1.02, 1.46)	0.03
Z62.9 Problem Related to Upbringing Unspecified	11 (0.00%)	2.56 (0.77, 8.55)	0.13
Z63.1 Problems in Relationship with In-Laws	14 (0.00%)	3.56 (1.21, 10.43)	0.02
Z63.5 Disruption of Family by Separation and Divorce	395 (0.07%)	1.34 (1.08, 1.66)	0.01
Z63.7x Other Stressful Life Events Affecting Family and Household	348 (0.06%)	1.24 (0.98, 1.57)	0.08
Log-Likelihood	3.076 × 10^5^		
LLR *p*-value	0.000		
AUC	0.6016		

Source: Authors’ analysis of IQVIA claim data between January 2019 and December 2021.

**Table 9 healthcare-13-02102-t009:** Adjust the odds ratio for the number of SDoH categories on 30-day hospital readmission for heart failure.

	N	Adjusted Odds Ratio (95% CI)	*p* Value
SDOH 1	29,308	1.23 (1.20, 1.26)	0.00
SDOH >= 2	2873	1.62 (1.50, 1.75)	0.00
Log-Likelihood		3.077 × 10^5^	
LLR *p*-value		0.000	
AUC		0.601	

Source: Authors’ analysis of IQVIA claim data between January 2019 and December 2021.

## Data Availability

Restrictions apply to the availability of these data: Data were obtained from IQVIA and are only available from the authors with the permission of IQVIA.

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
