# Peer review of "Social Determinants of Health and 30-Day Readmission for Heart Failure Patients in U.S. Hospitals: Evidence from ICD-10 Z-Code Data"

_healthcare, 2025, doi:10.3390/healthcare13172102_

Round 1
Reviewer 1 Report
Comments and Suggestions for Authors
Title : Social Determinants of Health and Hospital Readmission for Heart Failure Patients
This study highlights the usefulness of Z-codes as tools for identifying social risk factors, with a view to improving the management of patients with heart failure. It emphasizes the need for policymakers and healthcare professionals to incorporate social determinants of health (SDoH) into readmission prevention strategies and to design targeted interventions based on data related to social risk factors.
Introduction
- Line 35 : The paragraph mentions that only a small set of SDoH has been studied , but it would strengthen the rationale to name a few key SDoH that were typically included/excluded in previous research.
- Better specify the research question or hypothesis : Consider explicitly stating the research question or hypothesis at the end of the introduction, to clarify what exactly the study intends to test or explore.
- Minor language adjustments :Line 26: 'has been a focus on improving', revise to 'a focus of efforts to improve' for grammatical accuracy." and Line 48: ‘ICD_10 Z-codes’, ensure consistent formatting (ICD-10).
The introduction does a good job of contextualizing the clinical relevance of HF readmission and its costs.
Methods
The methods section provides a solid foundation with a large, nationally representative dataset and an appropriate use of claims and prescription data. However, several clarifications and improvements are recommended to enhance transparency and reproducibility :
- The rationale for excluding hospitals with below-median Z-code reporting rates is understandable, but it introduces a potential selection bias. A discussion of how this may affect generalizability would strengthen the methodology.
- The inclusion of CCI and demographic variables is appropriate, but social confounders outside Z-codes (e.g., insurance instability, mental health diagnoses) are not mentioned and could be relevant.
- The methodology mentions excluding patients with missing gender or birth year, but does not specify how other types of missing data were handled, particularly in diagnostic or Z-code variables.
- The use of p-values is noted, but a discussion of effect sizes and their clinical relevance (e.g., interpretation of odds ratios in context) would add depth to the findings.
Results :
The results section is well-organized and methodologically sound, with clear descriptions of patient baseline characteristics and appropriate use of Chi-square tests and multivariate logistic regression. The detailed analysis of SDoH sub-codes adds valuable insight into specific social determinants associated with 30-day readmission. However, including visual representations (e.g., forest plots) could enhance clarity.
Discussion :
The discussion in this article effectively highlights the growing interest in using ICD-10 Z-codes to better understand the impact of social determinants of health (SDoH) on clinical outcomes, particularly 30-day readmissions among patients with heart failure (HF).
However, the discussion is somewhat limited by the absence of qualitative analysis or contextualization of the results in real-world clinical practice. The comparison made with other studies (such as that of Sterling et al.) is relevant, but it could have been explored further to better understand the discrepancies in outcomes depending on the health indicators used (mortality vs. readmission). Moreover, the discussion could have placed greater emphasis on the persistent underutilization of Z-codes in clinical practice and proposed concrete strategies to improve their adoption.
Conclusion : OK
Reviewer 2 Report
Comments and Suggestions for Authors
Again the are troubles to send my comments.I sent it directly

Reviewer 3 Report
Comments and Suggestions for Authors
This retrospective observational study examines the link between ICD-10 Z-codes, which are used to systematically record Social Determinants of Health (SDoH), and 30-day hospital readmission rates among heart failure (HF) patients. Using claims data from IQVIA’s Charge Data Master and additional datasets, the authors analyzed over 586,000 patients hospitalized for HF between 2019 and 2021. They found that several categories and subcategories of SDoH Z-codes—especially those related to housing, employment, social environment, and primary support groups—were significantly linked to a higher risk of readmission. Patients with two or more SDoH Z-codes had a substantially increased risk of readmission compared to those with one or none.
The study uses a retrospective cohort design, which is suitable for the research question. The large, nationally representative dataset enhances the generalizability of the results. The three-year observation period, along with both inpatient and outpatient claims data, provides a strong longitudinal perspective.
Eligibility criteria are clearly defined, and excluding hospitals with low or inconsistent Z-code reporting is justified to prevent statistical bias. However, while this enhances internal validity, it may decrease external validity by limiting representation from institutions that underreport SDoH data.
The primary outcome, 30-day all-cause hospital readmission, is clinically important and clearly defined. SDoH exposures are classified using ICD-10 Z-codes and examined at both the category and sub-code levels, providing detailed detail.
The use of chi-square tests and multivariate logistic regression models is a methodologically sound approach. Covariates such as age, comorbidity index (CCI), insurance type, and hospital characteristics are appropriately included. The statistical reporting (ORs with 95% CI and p-values) is clear and suitable.
The study recognizes the following limitations:
- Underreporting of Z-codes: Because Z-code use is inconsistent and remains quite low, the study probably underestimates the actual prevalence of SDoH issues.
- Coding Variability: The same SDoH issue might be represented by multiple codes, and coding practices may vary among providers.
- Unmeasured Confounding: Although many demographic and clinical covariates are included, there may be unaccounted confounders affecting both SDoH and readmission risk.
These limitations are inherent to observational studies that use administrative data and are properly disclosed in the discussion.
The main findings are clearly presented:
- Housing instability, unemployment, social isolation, and family disruption greatly raise the likelihood of readmission.
- A clear dose-response relationship is observed between the number of SDoH codes recorded and the number of SDoH codes recorded.
The discussion effectively connects findings to existing research and suggests practical actions for healthcare providers and policymakers.
This is a well-executed and relevant study that effectively links clinical outcomes with social determinants through a structured coding framework. Minor revisions could improve the manuscript.
While the logistic regression models are properly used and odds ratios are clearly presented, the manuscript does not include any model fit diagnostics, such as the Hosmer–Lemeshow goodness-of-fit test, area under the curve (AUC), or pseudo-R² values. Adding these metrics would improve the statistical rigor and allow readers to evaluate how well the model distinguishes between patients who are readmitted and those who are not. Given the high dimensionality of the model (multiple covariates and sub-Z-codes), reporting these indicators is essential for demonstrating robustness and preventing overfitting.
The manuscript does not explicitly mention whether data imputation or handling of missing variables (such as for demographic or hospital characteristics) was performed. In large administrative datasets, even small amounts of missing data can introduce bias or impact the validity of multivariate models. Although the sample size is large, a brief explanation of how missing data was assessed and managed (for example, listwise deletion, imputation, exclusion criteria) would improve transparency and reproducibility.
Round 2
Reviewer 1 Report
Comments and Suggestions for Authors
I have now reviewed the integration of the comments provided by the authors for the manuscript entitled "Social Determinants of Health and 30-Day Readmission for Heart Failure Patients in U.S. Hospitals: Evidence from ICD-10 Z-Code Data."
I have no further comments on this revised and improved version of the manuscript.
Reviewer 2 Report
Comments and Suggestions for Authors
I am completely satisfied.The authors not only practically took into acount my comments and suggestions,but also made changes on their own initiative.In fact, a new paper has been created that adequately describes the problem,presents nthe result clearly and concludes correctly.It is fascinating how a completely different - much better,of course- trxt can be developed from the same research material.Although the description concrns USA, it is possible to find issues that could applid to other health system.
The paper meets now the requirements for publication in its presented form.